

# An empirical evaluation of link quality utilization in ETX routing for VANETs

Raad Al-Qassas[1] and Malik Qasaimeh[2]

[1] Princess Sumaya University for Technology, Amman, Jordan
[2] Jordan University of Science and Technology, Irbid, Jordan

## ABSTRACT

Routing in vehicular *ad hoc* networks (VANETs) enables vehicles to communicate for safety and non-safety applications. However, there are limitations in wireless communication that can degrade VANET performance, so it is crucial to optimize the operation of routing protocols to address this. Various routing protocols employed the expected transmission count (ETX) in their operation as one way to achieve the required efficiency and robustness. ETX is used to estimate link quality for improved route selection. While some studies have evaluated the utilization of ETX in specific protocols, they lack a comprehensive analysis across protocols under varied network conditions. This research provides a comprehensive comparative evaluation of ETX-based routing protocols for VANETs using the nomadic community mobility model. It covers a foundational routing protocol, *ad hoc* on-demand distance vector (AODV), as well as newer variants that utilize ETX, lightweight ETX (LETX), and power-based light reverse ETX (PLR-ETX), which are referred to herein as AODV-ETX, AODV-LETX, and AODV-PLR, respectively. The protocols are thoroughly analyzed *via* ns-3 simulations under different traffic and mobility scenarios. Our evaluation model considers five performance parameters including throughput, routing overhead, end-to-end delay, packet loss, and underutilization ratio. The analysis provides insight into designing robust and adaptive ETX routing for VANET to better serve emerging intelligent transportation system applications through a better understanding of protocol performance under different network conditions. The key findings show that ETX-optimized routing can provide significant performance enhancements in terms of end-to-end delay, throughput, routing overhead, packet loss and underutilization ratio. The extensive simulations demonstrated that AODV-PLR outperforms its counterparts AODV-ETX and AODV-LETX and the foundational AODV routing protocol across the performance metrics.

# INTRODUCTION

Over the past years, vehicular *ad hoc* networks (VANETs) have gained substantial attention for their immense potential range of applications, encompassing not only the enhancement of road safety but also the provision of traveler entertainment through the seamless exchange of real-time data among vehicles. This surge in interest can be primarily attributed to the notable progress in wireless communications and mobile computing,

Corresponding author
Raad Al-Qassas, raad@psut.edu.jo

which have enabled the seamless integration of VANETs with various networking components. Indeed, VANETs represent a vital constituent of the broader intelligent transportation systems (ITS) (*Su & Tong, 2023*), which aims to revolutionize the transportation infrastructure by offering an array of automated services. The advancements in wireless communications and mobile computing allowed the smooth integration of VANET with other networking entities.

In the context of VANETs, two distinct communication modes come into play: short-range and long-range. The short-range communication mode is characterized by vehicles communicating in an *ad hoc* fashion using emerging wireless technologies like 802.11p (*Gerla & Kleinrock, 2011*), recognized as vehicle-to-vehicle (V2V) communication (*Al-Sultan et al., 2014*). Alternatively, vehicles can benefit from road side units (RSUs) positioned along roadways, known as vehicle-to-infrastructure (V2I) communication (*Chatterjee et al., 2022*), which enables vehicles to access various information about road conditions, traffic updates, safety alerts, and local attractions. Figure 1 provides an example of VANET with moving vehicles and RSUs available on the road.

Vehicles can also utilize different technologies, including cellular networks, to support long-range communication. However, regardless of the communication range or method employed, transmitting data between nodes within a VANET necessitates the implementation of a routing protocol. This protocol is pivotal in establishing multi-hop paths between communicating nodes, ensuring the reliable and efficient flow of data within the network. VANET routing has several challenges: scalability, highly dynamic topology, link disruptions, environmental impact, and security. Routing can be a challenging task with the short time connectivity between entities caused by frequent link disruption, resulting in data loss. Routing protocols should incorporate scalability mechanisms to handle the variation in network density and allow the proper use of network resources.

The potential applications of VANETs can be neatly classified into two broad categories: safety and non-safety applications (*Engoulou et al., 2014*; *Gerla & Kleinrock, 2011*; *Lu, Qu & Liu, 2019*). Within the scope of safety applications, VANETs can play an influential role in critical areas such as traffic control, disaster recovery, and hazard prevention. They form the digital backbone of systems aimed at managing and optimizing traffic flow, swiftly responding and recovering to disasters, and proactively preventing potential hazard situations on the road. On the other hand, non-safety applications encompass a spectrum of services, including the dissemination of road service information and facilitating group communication among road users. In this context, VANETs serve as a means for sharing real-time updates on road conditions and services, which enrich the driving experience. Additionally, they enable efficient group communication, enabling various participants on the road to stay connected and collaborate seamlessly.

Various mobility models are available to demonstrate the nodes' mobility, which can be categorized as pure randomized mobility models, time-dependent mobility models, path-planned mobility models, group mobility models and topology-based mobility models (*Bujari et al., 2017*). Pure randomized mobility models, such as the random waypoint model (*Johnson & Maltz, 1996*), involve random movement in terms of direction, speed,

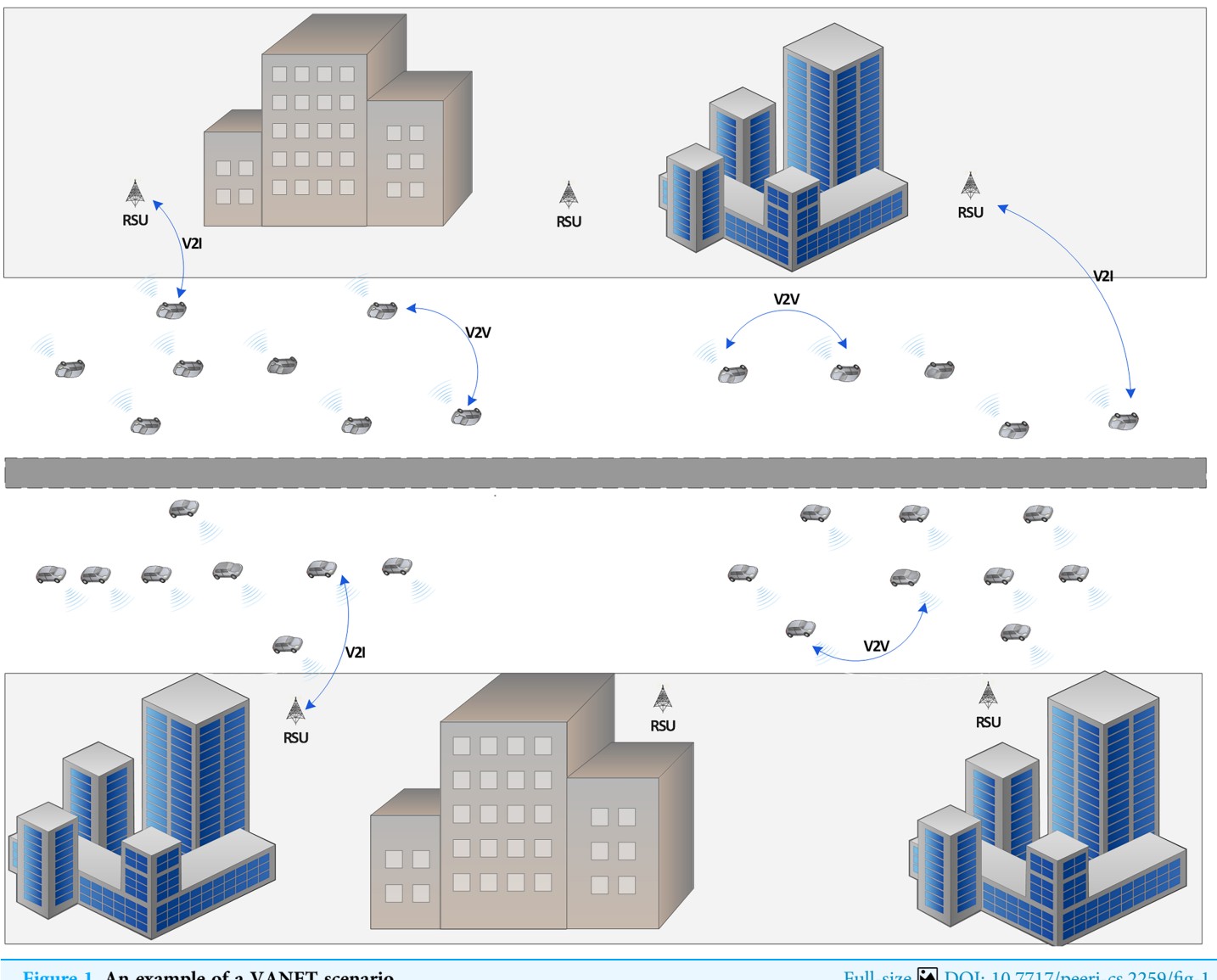

**Figure 1  An example of a VANET scenario.**       

and time. They are typically used to simulate nodes with purely independent movements.
Time-dependent mobility models such as the smooth turn mobility model (*Wan et al., 2013*) rely on moving nodes' previous speed and direction to ensure a smooth change of motion. Path-planned mobility models, such as the paparazzi mobility model (*Bouachir et al., 2014*), deploy a certain predefined path in order to force the nodes to follow it where nodes follow a certain predefined path, without taking any random direction. Group mobility models where node movements are constrained by a reference point for each group such as the nomadic community mobility model (*Bujari et al., 2017*) and reference point group mobility model (*Al-Qassas & Qasaimeh, 2022*). The nodes move randomly within a specific area around the reference point. Group mobility models include a spatial constraint among mobile nodes within a group. Finally, topology-based

mobility models (*Messous, Senouci & Sedjelmaci, 2016*), nodes move around and are aware of their topology, coordinating their positions with each other. This is typical in mission-constrained environments where nodes must coordinate their positions and stay connected during a specific task. These mobility models are designed to maintain a fully connected network at all times. Such mobility models are common in flying *ad hoc* networks (FANETs), as they facilitate network topology control and data transmission.

Due to the limitations in wireless communication in VANET, and in order to improve the selection quality of communication links forming a path between communicating nodes, there has been a demand to propose enhancements on routing protocols designed for VANET. One of the major enhancements is the utilization of the expected transmission count (ETX) to estimate link quality. ETX is utilized in the routing protocols to enhance their operation (*Liu et al., 2017*; *Malnar & Jevtic, 2022*; *Septa & Wagh, 2023*; *Smida, Fantar & Youssef, 2018*; *Son et al., 2014*; *Xie, 2018*). ETX captures factors like packet loss, collisions, and link asymmetry, providing a cost metric proportional to the resources needed to transmit data over a route. Depending on the robust estimation of link quality using ETX, routes with lower cumulative ETX typically could provide better throughput and the needed efficiency required by the routing protocol. As an enhancement on routing protocols for VANET, ETX has been employed within widely studied VANET routing protocols such as the well-known AODV, OLSR and geographical routing protocols (*Gangopadhyay & Jain, 2023*; *Malnar & Jevtic, 2022*; *Septa & Wagh, 2023*; *Xie, 2018*) as a core link cost metric to provide enhanced protocols, and have been incorporated as a key metric for optimal route selection based on link quality and reliability. These protocols highlight the potential of ETX advancements to improve efficiency, delay, overhead, and stability performance in VANET routing. However, most existing research works have studied protocols or techniques in isolation under limited communication settings (*Ardianto et al., 2022*; *Bugarcic, Malnar & Jevtic, 2019*; *Jevtić & Bugarčić, 2023*; *Malnar & Jevtic, 2022*; *Malnar & Jevtić, 2020*; *Purnomo et al., 2018*; *Smida, Fantar & Youssef, 2018*). These studies lack the comprehensive analysis to consider the breadth of ETX optimization approaches, realistic mobility models and communication traffic conditions. This has constrained insights into selecting appropriate ETX-based routing protocols for real-world VANET deployment supporting safety and non-safety applications.

To address this gap, this research undertakes an extensive comparative evaluation of diverse ETX-based routing protocols for VANETs encompassing both a foundational routing protocol and newer variants leveraging ETX optimizations. These protocols are systematically analyzed *via* simulations under varied network conditions. Nodes in VANETs may collaborate with each other and might behave in a nomadic way (*Barnard, Fischer & Flament, 2015*; *Chriki et al., 2019*; *Lin et al., 2020*; *Quan et al., 2014*; *Sáez et al., 2009*; *Tadros et al., 2021*; *Zang et al., 2008*). To investigate this kind of movement behavior, the study employs the nomadic community mobility model (*Bujari et al., 2017*), which can represent this kind of movement behavior where it provides the movement scenarios of

groups of vehicles that may move collectively but independently within a practical range. Our three core contributions are:

(i) Evaluating the impact of varying traffic conditions and providing analysis of efficiency, overhead, delay, and stability of renowned ETX-based protocols.

(ii) Evaluating the impact of the nomadic mobility model on ETX optimization routing techniques.

(iii) Providing an insight into designing robust, adaptive ETX-based VANET routing solutions for emerging ITS applications through a better understanding of their performance under different working conditions.

For this purpose, we consider the well-known *Ad hoc* On-demand Distance Vector (AODV) (*Perkins, Belding-Royer & Das, 2003*) routing protocol and three ETX routing protocols, namely AODV-ETX (*Couto et al., 2005*), AODV-LETX (*Jevtic & Malnar, 2019*), and AODV-PLR (*Malnar & Jevtic, 2022*). The analysis under varied environments offers helpful guidelines for implementing ETX-based routing in future VANET networks. Further, it will provide broader insights into utilizing ETX advancements tailored to vehicular environments beyond isolated protocol specific studies.

The rest of this article is structured as follows. "Literature Review" conducts a comprehensive literature review, delving into various routing protocols that incorporate the ETX metric in their operation. "*Ad hoc* On-demand Distance Vector (AODV)" provides an extensive explanation of AODV routing protocol. "AODV-ETX" explores the details of the AODV-ETX routing protocol. "AODV Light ETX (AODV-LETX)" provides a thorough illustration of the AODV-LETX operation. "AODV Power-Based Light Reverse ETX (AODV-PLR)" explores the ETX unique characteristics of the AODV-PLR protocol. "Nomadic Community Mobility Model" overviews the nomadic mobility model and explains its characteristics. "Simulation Environment" comprehensively describes the simulation environment used in the study, including related settings and configurations. "Evaluation" explains the performance metrics and provides a detailed analysis of the performance results, providing insights into the effectiveness of the various protocols. Finally, in "Conclusion and Future Work", the study concludes its findings and outlines potential directions for future work.

## LITERATURE REVIEW

ETX has emerged as a promising enhancement for routing protocols in VANETs. Various routing protocols have incorporated ETX as a core metric for optimal route selection based on link quality and reliability estimates, such as AODV-ETX, AODV-LETX, AODV-PLR, GEO-LU (*Alzamzami & Mahgoub, 2020*), LSGO (*Cai et al., 2014*), SCAOR (*Sadatpour, Zargari & Ghanbari, 2019*), F-ETX (*Bindel, Chaumette & Hilt, 2016*), DQLTV (*Smida, Fantar & Youssef, 2018*), SD-TDQL (*Zhang et al., 2022*). Researchers have developed enhanced versions of the popular protocols like AODV, OLSR, and geographical routing (*Gangopadhyay & Jain, 2023*; *Jevtic & Malnar, 2019*; *Malnar & Jevtic, 2022*; *Septa & Wagh, 2023*) that significantly improve routing protocol performance and can potentially reduce delays, overhead, and instability through more innovative path selection that adapts to

dynamic network conditions. The integration of ETX with well-known routing protocols proves its viability as an impactful advancement for building efficient and resilient routing protocols. Table 1 provides a detailed comparison between these methods with a focus on five measures, including the protocol performance focus, its utilized parameters, the information exchange frequency needed in its operation, the path selection approach, and identified drawbacks.

Link state aware geographic opportunistic (LSGO) (*Cai et al., 2014*) combines geographic location and the link state in the implementation of ETX metric to predict reliable paths and reduce packet loss caused by mobility. In LSGO, each node periodically broadcasts hello packets to estimate link quality. To send data, the node selects a set of candidate forwarders based on neighbor locations and link ETX values. The sender assigns priorities to candidates based on the distance to the destination and ETX; the closer nodes with lower ETX are given higher priority. The candidate list and priorities are included in the packet header. The highest priority candidate transmits first when it receives the packet, while lower priority candidates wait, with the aid of timers, for higher priority ones to transmit the packet. If a timer expires without hearing a transmission, it transmits the packet. This is called opportunistic forwarding *via* candidate set, where multiple potential forwarders are used to ensure packet transmission toward the destination despite link failures. However, this approach may result in increasing overhead and may result in delays while nodes wait for higher-priority candidates to transmit. It also requires periodic Hello packets to estimate link quality, incurring additional overhead. Furthermore, as network density increases, the size of the candidate sets may increase, resulting in an increase in packet header size.

Collision Aware Opportunistic Routing (SCAOR) (*Sadatpour, Zargari & Ghanbari, 2019*) is an opportunistic routing protocol. It uses location info to spatially separate contention areas to reduce collisions resulting from hidden nodes. It utilizes three parameters, including link quality, node density, and packet advancement. Each node maintains a table of its neighbors with their positions and collision domains. The neighbor's information is exchanged *via* hello packets. In order to forward the received packet to its destination, the neighbor with the lowest collision probability is selected for this purpose. The collision probability for each neighbor is calculated based on their collision domains. If all neighbors have a high collision probability, it simply switches to perimeter routing. The idea of this protocol is to avoid collisions and improve the packet delivery ratio compared to greedy geographical routing protocols. However, additional overhead results from exchanging neighbor collision domain information. Also, the calculation of collision probabilities may result in processing complexity. Furthermore, it does not consider link quality, and the effectiveness of its operation depends on the accuracy of collision domain estimation.

F-ETX (*Bindel, Chaumette & Hilt, 2016*) employs ETX in its operation. It estimates the expected number of transmissions needed for a packet to be successfully delivered from source to destination. The estimation accounts for loss ratios in both directions of a link. The forwarding load and channel diversity on each path are considered in the calculation. The main target for F-ETX is to improve load balancing and reliability. It also adds a

**Table 1 A detailed comparison between ETX methods.**

| Method | Performance focus | Utilized parameters | Exchange frequency | Path selection approach | Drawback |
|---|---|---|---|---|---|
| LSGO (*Cai et al., 2014*) | • Reliable paths<br>• Mobility resilience | • Distance to destination<br>• Link state | Periodic | • Opportunistic forwarding *via* candidate set<br>• Geographic routing | • Increased overhead<br>• Delay waiting for higher priority candidates<br>• Candidate sets may increase |
| SCAOR (*Sadatpour, Zargari & Ghanbari, 2019*) | • Collision avoidance<br>• Node density adaptation | • Link quality<br>• Node density<br>• Packet advancement | Periodic | • Opportunistic, through the neighbor with the lowest collision probability | • Calculation of collision probability<br>• Collision domain information exchange |
| F-ETX (*Bindel, Chaumette & Hilt, 2016*) | • Load balancing<br>• Reliability<br>• Channel diversity | • Loss ratios<br>• Forwarding load<br>• Channel diversity | Periodic | • Lowest ETX | • Increased computation complexity |
| Fuzzy F-ETX (*Septa & Wagh, 2023*) | • Contention window optimization<br>• Node density adaptation | • F-ETX<br>• Coverage factor<br>• Direction<br>• Velocity difference | Periodic | • Fuzzy logic relay selection | • Increased computation complexity<br>• Communication overhead |
| DQLTV (*Smida, Fantar & Youssef, 2018*) | • Video streaming<br>• Improve link quality<br>• Maximize link lifetime | • Link quality<br>• Link lifetime<br>• Delay<br>• Hop count | Periodic | • Route discovery based on AODV | • Frequent beaconing<br>• Communication overhead from local path updates |
| Geo-LU (*Alzamzami & Mahgoub, 2020*) | • Traffic load adaptation<br>• Link conditions | • Link quality<br>• Bandwidth<br>• Distance progress | Periodic | • Opportunistic forwarding Localized two-hop neighbor selection<br>• Geographic routing | • Bandwidth estimation can be challenged by hidden terminal problem<br>• Two-hop neighbor discovery |
| SD-TDQL (*Zhang et al., 2022*) | • Optimized end-to-end ETX<br>• Trust model adaptivity | • Link quality<br>• Trust value | Periodic | • Deep Q-Learning path selection | • Centralized calculation with global knowledge, which yields optimized end-to-end path ETX<br>• Learning complexity<br>• Trust model can affect performance |
| AODV-ETX (*Couto et al., 2005*) | • Routes quality | • Link loss ratio | On-demand | • AODV-based path selection | • Header size |
| AODV-LETX (*Jevtic & Malnar, 2019*) | • Routes quality<br>• Reduce the ETX field length | • Optimized link loss ratio | On-demand | • AODV-based path selection | • Coefficient value determination |
| AODV-PLR (*Malnar & Jevtic, 2022*) | • Routes quality<br>• Reduce overhead | • Packet Loss Ratio<br>• RSSI influence | On-demand | • AODV-based path selection | • Coefficient value determination |

forwarding load factor to the ETX calculation, representing an estimation of how many other flows are forwarded by a node. A channel diversity factor is also included to allow diverse use of channels. Including all these factors in the calculation of ETX, F-ETX may provide better load distribution across links and uses paths with better channel diversity. However, including all these factors results in an increased computation complexity.

*Septa & Wagh (2023)* proposed an enhancement on F-ETX. The enhancements utilize fuzzy logic in two components: the relay selection and the optimization of contention window size. For relay selection, each node periodically broadcasts hello packets to exchange information like location and direction. Nodes use the exchanged information to compute F-ETX, coverage factor, direction, and velocity difference, for the neighboring nodes. These four components are injected into the fuzzy logic system to select the best next-hop relay node. The contention window optimization is adaptive and depends on three factors, including velocity, density, and link quality to the selected relay. These factors are injected into another fuzzy logic system to determine the optimal window size. This adaptability helps to reduce collisions, which can improve network throughput. However, it requires additional computation and communication overhead, and it is dependent on keeping accurate information at each node. Lack of accuracy in the exchanged information may result in inaccurate judgment produced by fuzzy logic systems.

*Smida, Fantar & Youssef (2018)* proposed a routing protocol named DQLTV for video streaming in VANETs that considers delay, link quality, and link lifetime. The protocol uses a mathematical model to improve link quality and maximize its lifetime while maintaining low delay when forming the forwarding paths. The link quality is measured based on the probabilities of successful packet transmission and acknowledgment between nodes by counting sent and received packets. The link lifetime is calculated based on the distance between nodes and direction using turn signal data. The route discovery phase is based on AODV with additional fields added to include the needed information, including distance, link quality, and link lifetime. These details are updated at each hop. Of course, multiple paths can be discovered, in which case the best path is chosen based on a forwarding function that takes into consideration the four factors, including link quality, link lifetime, delay, and hop count. If the link lifetime of the next hop falls below a specified threshold, a local path updating is conducted, and a new RREQ is sent to find an alternate next hop based on the forwarding function. Although ETX itself is not used directly as a metric in DQLTV, the probabilities used to calculate link quality are based on a similar concept. Local repairs minimize path failures, and video buffering enables faster access if nearby vehicles have video transmission. However, routing overhead may result from frequent beaconing and local path updates.

*Alzamzami & Mahgoub (2020)* proposed a greedy protocol named Geo-LU that selects the best pair of subsequent two-hop nodes pair from the forwarding node based on three factors, including distance progress in the direction to the destination, available bandwidth, and link quality. Geo-LU does not select the complete end-to-end path

between source to destination; it makes localized routing decisions at each hop to select the next forwarding node pair consisting of one and two-hop neighbors. This is to minimize the overhead of computing a whole path to the destination. The use of two-hop information allows better routing decisions compared to protocols that use only one-hop neighbors. Each node collects various details about its one and two hop neighbors, including their positions, velocities, headings, link quality metrics, residual bandwidth, and consumed bandwidth, which is maintained in a table designed for this purpose. These details are exchanged periodically using hello packets. Packet forwarding begins by obtaining the destination location position and then the process of selecting the candidate next-hop pairs. For each pair, the protocol predicts their current positions based on data history. The cost for selecting each pair is calculated based on the link utility (LU) that considers the minimum outstanding bandwidth on the link and the link quality measured by ETX and distance progress to the destination. The outstanding bandwidth is estimated based on the data rates of nodes. Geo-LU is considered adaptive to changes in network traffic load and link conditions. However, it requires more overhead resulting from the two-hop neighbor discovery using large hello packets, in addition to its computation complexity in evaluating the two-hop pairs at each hop compared to one-hop greedy forwarding. Also, bandwidth estimation can be challenged by hidden terminal problem. Furthermore, it uses a simple position prediction model to lower the complexity.

Zhang et al. (2022) proposed a software-defined framework named SD-TDQL that utilizes ETX to estimate link quality. It uses the forwarding ratio of each node as the probability of successful forward transmission as trustworthiness probability. Each node calculates the trust value of its neighbors based on packet forwarding trustworthiness. Nodes exchange trust values. The SDN controller collects the trust values and delivery ratios to calculate the ETX for each potential link. The framework uses deep Q-learning to determine which link minimizes ETX and maximizes long-term reward. The framework provides a precise link quality metric for the SDN controller to optimize through deep reinforcement learning, which allows fine-grained link quality differentiation. The use of SDN allows centralized calculation with global knowledge, which yields optimized end-to-end path ETX. The use of deep Q-learning optimizes ETX, allows adaptivity, and improves performance. However, centralized control has a single point of failure, which can be managed with controller redundancy. Also, it is essential to note that deep reinforcement of learning complexity requires significant training time. Furthermore, the trust model can affect performance.

## AD HOC ON-DEMAND DISTANCE VECTOR

AODV (Perkins, Belding-Royer & Das, 2003) is a well-known representative of reactive routing protocols designed for dynamic wireless networks. It creates routes only when a node initiates data transmission. The design of this protocol was to avoid the unnecessary communications needed to maintain routes regardless of data transmission demands, with the aim of efficient resource utilization. AODV achieves this by employing a route

discovery mechanism that substantially reduces the necessary flooding operations when seeking a new route and route maintenance to handle the broken routes between two nodes.

When data transmission is commenced from a source node to a specific destination, a route discovery phase is triggered to determine a communication path between them. The route discovery is initiated by broadcasting a *route request* (RREQ) packet from the source node to its immediate neighbors. Each node will forward the RREQ to its immediate neighboring nodes, thus cascading through the network. Of course, each node keeps track of these RREQ packets in its record, so in case this RREQ has been received before, it will be broadcasted only once with the first RREQ. For this purpose, the RREQ has a unique identifier to allow this process to be achieved in the proper manner. The forwarding of RREQ packets continues until the RREQ successfully reaches its intended destination node. Importantly, each node that participates in the forwarding process of RREQ adds a reverse route in its routing table to the source node that initiated the RREQ, which contributes later to building the path and will be used by the route reply. Notably, due to the broadcast nature of RREQ, duplicates received by intermediate nodes between the source and destination are discarded to avoid unnecessary redundancy.

Upon receiving the RREQ by the destination, a *route reply* (RREP) packet, which is a unicast packet sent back to the source node. This RREP packet traverses along the reverse route previously established as the RREQ propagated through the network. As this RREP traverses toward the source node, the participating nodes update their routing table entries with a route toward the destination node. It is important to emphasize that nodes forming this path do not require awareness of the complete path itself but retain the next hop to the destination, which indeed enhances the protocol efficiency and scalability. It is essential to note that in case there is more than one path available, the one with lower hop count is selected.

AODV relies on utilizing *sequence numbers* to guarantee fresh route information. It is highly important to maintain up-to-date routes and to avoid loops. Therefore, the destination's sequence number is essential in serving this purpose. Initially, the destination's last-known sequence number is embedded within the RREQ by the source node. An intermediate node can promptly respond to the RREQ if it has an up-to-date route to the destination, which can be recognized if the sequence number it has is higher than the one in the RREQ packet. This helps significantly in enhancing the operation of the network as it operates with accurate and timely routing information.

The route maintenance phase is initiated by the intermediary node in the established route to the destination upon discovering a broken link in the active route. This could be when the next hop node to the destination becomes unreachable. Upon this, the node removes from its routing table the associated entry and informs the affected neighboring nodes of the broken route by sending a *route error* to them. For this purpose, AODV utilizes an active neighbors list, which helps in tracking the active neighboring nodes that use a specific route. The route error keeps propagating toward the affected source nodes to inform them about the route break, which triggers a new route discovery phase if there is still data to be transmitted.

## AODV-ETX

In AODV-ETX (*Couto et al., 2005*), ETX is a metric to measure link quality embedded within the routing protocol operation to provide better quality routes. The link ETX represents the predicted data transmissions needed to transmit a packet over a link, which includes the retransmissions. It is calculated based on the successful probability of packet transmission ($p_f$) and the likelihood of receiving an ACK packet ($p_r$) successfully. The ETX for a link $l$ is calculated using Eq. (1).

$$ETX_l = \frac{1}{p_f \times p_r}.$$ 

(1)

The measurement of these probabilities is done using link probe packets (LPPs) broadcasted by every node. The LPP packets are broadcasted periodically over an average period $\tau$. No acknowledgment is required in order to reduce the overhead associated with the broadcast of LPP packets, although it has a fixed size. In order to avoid synchronization, a jitter of up to ±10% is used for the period $\tau$. Each node counts the received LPPs to compute the probability $p_r$ at time $t$, which is calculated according to Eq. (2), where $LLPs_r$ [t-w, t] represents the count of received LPP packets during the time window frame $w$, and $LLPs_a$ [t-$\tau$, t] corresponds to the anticipated LPPs to be received within period $\tau$.

$$p_r(t) = \frac{LPPs_r\ [\mathrm{t} - w,\quad \mathrm{t}]}{LPPs_a\ [\mathrm{t} - \tau,\quad \mathrm{t}]}.$$ 

(2)

To illustrate this, if we have a link between two nodes, let us say $x$ and $y$, the algorithm can compute $p_r$ by counting LPPs successfully received from y. However, considering that LPP acknowledgment is not required, $p_f$ cannot be concluded by x. To handle this, node y includes, with each LPP packet sent, the count of node x LPP packets received during the last $w$ seconds to allow the computation of $p_f$. The typical value for w is set to $w = 10\ \tau$, and the typical value of $\tau$ is set to 1 s. The ETX values of the links forming the route will determine the ETX value for the route, as illustrated in Eq. (3), where $ETX_l$ is the ETX for link $l$ within the route, and $ETX_r$ corresponds to the route ETX:

$$ETX route = \sum\nolimits_{l \in \text{route}} ETX_l.$$ 

(3)

The ETX measures, with the aid of LPP packets, the link loss ratio. Each node broadcasts an LPP packet periodically, typically every second, and maintains the details of received LPP packets from its neighbors over the past 10 s. The LPP packet includes the following fields: *type*, *ID*, *sender IP address*, *sender sequence number*, *neighbor IP address*, *forward LPP count*, and *neighbor count*. The neighbor IP address corresponds to the neighboring node from which the node received at least an LPP packet in the last 10 s. The forward LPP count indicates the received LPP packets count in the last 10 s from the neighbor with the IP address given in the previous field. These two fields are repeated for each neighbor.

Each node has a table with three entries associated with each one of its neighbors to track the details of each neighbor and to calculate the ETX. These entries include the *neighbor IP address, forward LPP count*, and *reverse LPP count*. The forward LPP count is to count received LPP packets from a node in the last 10 s. The reverse LPP count tracks the count of received LPP packets from its neighbor in the past 10 s.

AODV-ETX represents an enhanced routing protocol that incorporates ETX in its operation based on AODV (*Ardianto et al., 2022*; *Dubey & Dubey, 2014*). For this purpose, the RREQ and RREP packets include a field to represent ETX value. The ETX value in the initiated RREQ is set to zero. The ETX field in the RREQ is updated at each node in respond to Eq. (3) as the RREQ is propagated toward the destination. Similarly, RREP packets are processed in a similar way. In order to incorporate ETX, the routing table entry has a field for this purpose. Routes are classified based on their ETX value; the best route is the one with the smallest ETX value. However, in case there are multiple routes with the same smallest ETX value, the selection is based on number of hops.

When a source node needs to transmit data to a destination with no available route, it initiates a RREQ with the hop count and ETX fields initialized with zero. Intermediary nodes that receive the RREQ from a neighboring node will calculate the ETX value to the neighbor based on the previously obtained forward LPP count and reverse LPP count values. Then, it updates the ETX value of the source node in its routing table by summing the value ETX in the received RREQ and the computed ETX value to the neighboring node. Based on the status of the RREQ, it will be updated. If the status is 1, it updates with the calculated ETX and increments the hop count. Otherwise, it uses the current ETX value in the routing table and increments the hop count.

Unlike original AODV, intermediate nodes handle all RREQ packets from already-seen RREQ packets from the same source node. The RREQ is rebroadcasted in one of the two cases, if the ETX value in the received RREQ is smaller than the value in the routing table or in the case that ETX values are the same and the RREQ has a lower hop count; otherwise, the RREQ is discarded. Nonetheless, when the destination receives the RREQ, it updates the routing table entry associated with the source by updating the next hop to the destination and the ETX value. Then, the destination generates a RREP packet with both hop count and ETX fields initialized with zero and sends it through the reverse path.

When an intermediate node receives a RREP packet, it updates the hop count and the ETX value, and it updates the routing table entry to the destination node. The ETX value is updated by adding the ETX previous link. The updated RREP traverses toward the source node over the best ETX route. All intermediate nodes repeat this procedure until the RREP reaches the source node, which will make all needed updates and starts transmitting data packets. However, in case additional RREP packets are received later with better ETX, the routing table is updated accordingly and used in transmitting data packets.

## AODV LIGHT ETX

AODV-LETX (*Jevtic & Malnar, 2019*) added some enhancements over AODV-ETX. Among the enhancements introduced in AODV-LETX is the reduction of the details

included in the routing packets that are needed to calculate the ETX field to reduce the packet size. Based on the value of $p_r$ or $p_f$, which represent the packet reception probabilities for the link, the original ETX for link $l$ within the route calculated using Eq. (4) has a value that can range from 1 to infinity, with a value equal to 1 when both probabilities are equal to 1 and infinity when at least one of the probabilities equals zero. Values follow a highly non-uniform distribution, which means that values are likely to be around 1, while large values are infrequent, and hence ETX value is represented as a floating point in the packet header, which results in using four bytes for each ETX value, which is utilized in the RREQ and RREP packets, which can impact network performance considering the limited bandwidth.

$$ETX_l = \frac{1}{p_f \times p_r}. \tag{4}$$

A logarithmic function is used to deal with the values of non-uniform distribution to make the ETX field shrink to 1 byte. The idea of the function is to expand the differentiations between values. After applying the logarithmic function in Eq. (5), the resulting ETX value is rounded, converted to an unsigned integer value, and represented using only 1 byte. This means that the light ETX (L_ETX) metric of link $l$ can be between 0 and 255, with the smallest value corresponding to the best link and the highest value corresponding to the worst link.

$$L\_ETX_l = \lceil 60 \times \log(ETX_l) \rceil. \tag{5}$$

Reducing the ETX field length is crucial to minimize the impact on packet sizes since the already reserved bits available in the RREQ packet header can accommodate the 1-byte L-ETX metric. This means the overall RREQ packet size is the same as the original AODV, hence avoiding the routing overhead associated with increasing packet size. While the RREQ packet remains the same in size, RREP packets use an additional field to include the L-ETX metric. However, the impact of this additional field in RREP packets is not expected to significantly increase routing overhead since RREPs are unicast packets.

The coefficient value of 60 in Eq. (5) is not derived mathematically but is determined through empirical experimentation based on a trade-off between boosting the distinction between closely spaced ETX values and avoiding saturation for poor-quality links. The choice of this coefficient is critical because it influences how the L-ETX behaves. Lower coefficient values reduce the boost effect introduced by the logarithmic function for small values of $ETX_l$. A lower coefficient mitigates the impact of the logarithmic function and can lead to decreased performance. Conversely, larger values of the coefficient can introduce saturation problems, particularly in the presence of multi-hop low-quality links. Saturation typically refers to a point where the metric is maximized, and further increases do not accurately reflect the link quality. Therefore, larger coefficient values may not effectively differentiate between various levels of link quality for low-quality links.

## AODV POWER-BASED LIGHT REVERSE ETX

AODV-PLR relies on the power-based Light Reverse ETX metric, known as PLR-ETX, which is an enhancement of Light Reverse ETX (LR-ETX) (*Malnar & Jevtic, 2022*). In this study, we refer to the AODV PLR-ETX and AODV-PLR interchangeably. The enhancements are intended to lower the overhead associated with calculating the ETX metric, particularly in the cases of high-density networks, which can result in a large number of neighbors. LR-ETX is useful for links where the reverse probability $p_r$ is similar to the forward probability $p_f$. In general, ETX relies on both reverse probability $p_r$ and forward probability $p_f$ for link quality estimation. However, calculating $p_f$ can be challenging because it requires acknowledgments, often unavailable in many wireless networking scenarios. To compute $p_r$, each LPP packet includes the successfully received packet count by neighboring nodes within the past w seconds. This information can increase the size of LPP packets as the number of neighbors grows. In cases where the reverse probability $p_r$ is approximately equal to the forward probability $p_f$ (i.e., for mostly symmetric links), it is unnecessary to perform the calculations for $p_f$. Symmetric links are those where data transmitted in one direction is likely to be received successfully in the other direction. LR-ETX metric is designed for links where the calculation of $p_f$ is not required because of their symmetry, which is calculated according to Eq. (6) where $ETX_{rl}$ is ETX of the reverse link based on $p_r$, which is the successful reception of LPP.

$$ETX_{rl} = \frac{1}{p_r}. \tag{6}$$

In the context of ETX, LLPs are broadcasted periodically, typically every $\tau$ seconds, and counted within a time window frame of $w$ seconds to determine link quality. The values of $\tau$ and $w$ are determined experimentally; typically, $w$ is set to 10 $\tau$, with $\tau$ often set to 1 s. However, a value of 10 $\tau$ for $w$ could be large and may cause the ETX metric to respond slowly to changes in link quality. Decreasing w for a faster response is an option, but it reduces the resolution of the metric. On the other hand, decreasing $\tau$ to increase resolution would also increase overhead.

As a solution for enhancing the dynamic response, the metric uses the Received Signal Strength Indicator (RSSI), which measures the strength of the received signal, to improve the metric's responsiveness without significantly increasing overhead. This is defined in Eq. (7), where $ETX_{rl}$ is the reverse link ETX calculated using Eq. (6), and an $RSSI_{influence}$ is an influence factor defined in Eq. (8). The equation is scaled with coefficients 60 and 0.005 and uses a logarithmic function to compute the PLR-ETX metric of link $l$, which have been chosen experimentally. They are not derived from theoretical calculations but are determined through the analysis of simulations with the goal of achieving optimal network performance.

$$PLR\_ETX_l = \left[ 60 \times \log \left( ETX_{rl} \times \left( 1 + 0.005 \times RSSI_{influence} \right) \right) \right] \tag{7}$$

$$RSSI_{influence} = \begin{cases} 0, & RSSI \geq 0 \text{ dBm} \\ |RSSI|, & -100 \text{ dBm} < RSSI < 0 \text{ dBm} \\ 100, & RSSI \leq -100 \text{ dBm} \end{cases} \tag{8}$$

## NOMADIC COMMUNITY MOBILITY MODEL

The Nomadic Community Mobility Model (*Bujari et al., 2017*) is a group mobility model used to model the movement of groups of mobile wireless nodes that move collectively from one point to another. Nodes belonging to a group follow an invisible reference point; however, they can move freely within a predefined roaming radius. Each node in these groups has its own space to move around randomly. An interesting behavior in this model is that the reference point changes; hence, vehicles in each group will travel individually and randomly toward their new location within a predefined roaming radius of the reference point. This can be used in various situations, for example, a fleet of vehicles exploring a city, the vehicles move together from one location to another, but each vehicle explores specific areas on its own.

Within the nomadic community mobility model, there is a set of parameters that can describe the nomadic behavior, including an average number of nodes per group, maximum distance to the group center, group size standard deviation and reference point maximum pause. The average group size is random but controlled with the average number of nodes per group parameter. The group size standard deviation parameter is used to control how to vary the sizes of different groups from the average size. Group sizes are picked from a normal distribution with the mean. The standard deviation determines how much spread there is in the group sizes. A low deviation means that group sizes will be clustered closely around average size. A high deviation means that there will be more variation in group sizes, some much smaller or larger than the average group size. Typically, it could be 5–10 for modest variation.

The maximum distance to the group center is used to indicate the distribution radius away from the group center. This means that nodes within each group will be distributed around the group center, and no node will be located more than that radius away from the center. Nodes' traveling positions within this range will follow a specific distribution pattern. The reference point maximum pause parameter is used to set the association time of group members with the reference point. The association time with a reference point can be up to 60 seconds. This means that nodes will stay at a reference point for a maximum of 60 seconds before moving to another reference point.

All the settings mentioned above collectively define how nodes are grouped, how they move within those groups, the variability in group sizes, and how long they remain associated with reference points before continuing their movement. Figure 2 provides a simple illustration of this model with various groups of vehicles formed. Please note that there are no restrictions on the communication between vehicles from different groups or within the group itself.

## SIMULATION ENVIRONMENT

### Simulation model

The simulation experiments are conducted using the well-known ns-3 simulator, an established simulation tool extensively used in simulating wireless networks (*Campanile et al., 2020*). AODV, AODV-ETX, AODV-LETX, and AODV-PLR are implemented and

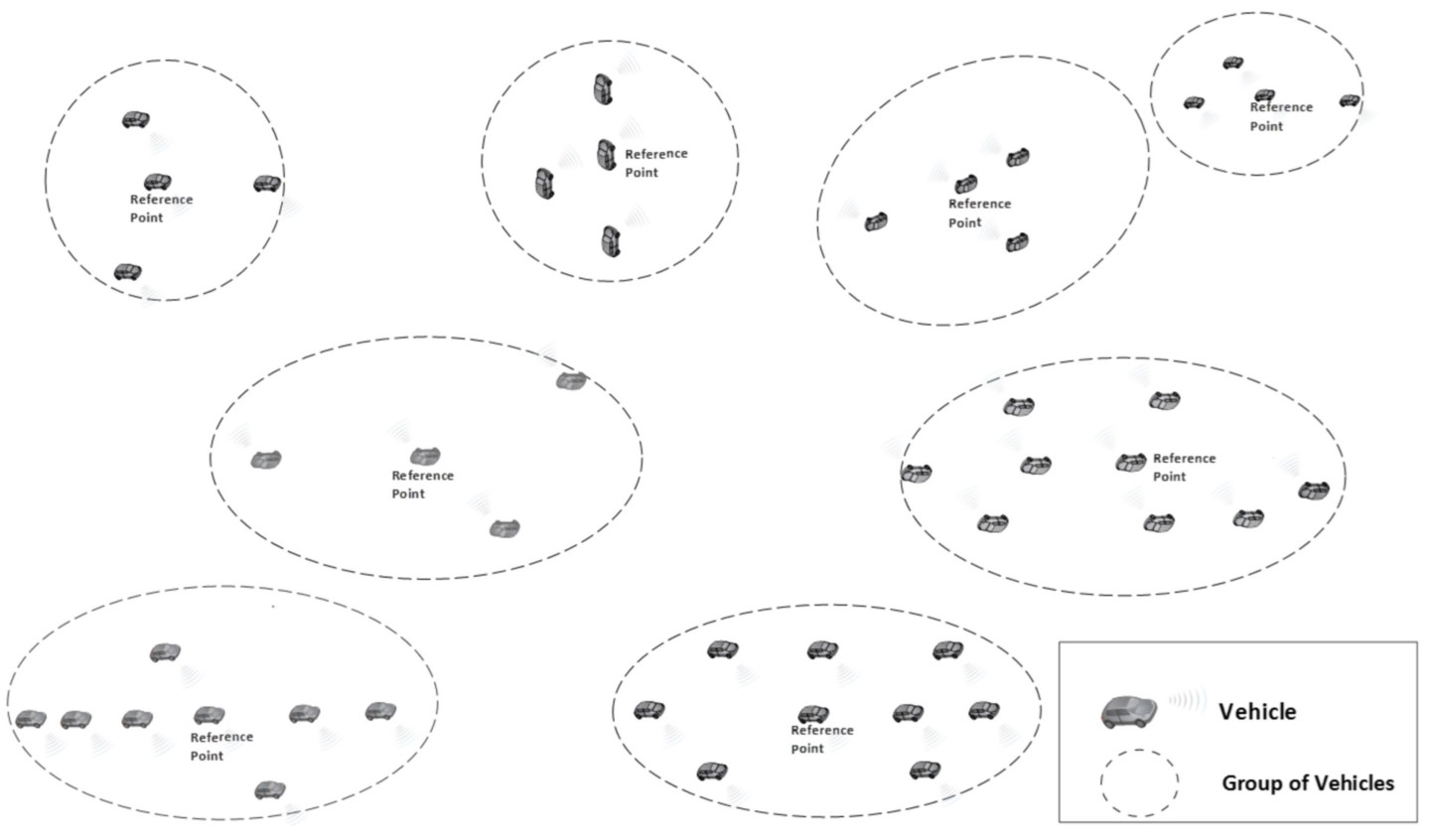

**Figure 2** **Nomadic mobility model illustration.**               

validated within the ns-3 code. The simulation model includes forming a network of wireless nodes and allows nodes to move over the simulated topology. This is governed by the generated VANET mobility scenarios that are built based on the following settings: simulation area, simulation time, mobility model, number of nodes, and average node speed. The node's movement follows the Nomadic mobility model described in "Nomadic Community Mobility Model". The scenario depicts the distribution of nodes and their movement across the simulation area. The mobility model governs the nodes' movement throughout the simulated time. These scenarios are created with the aid of the well-known Bonnmotion tool (*Aschenbruck et al., 2010*). In all simulated scenarios, nodes are equipped with the wireless standard IEEE 802.11 p (*Gerla & Kleinrock, 2011*), designed for VANET communications. The simulated networks have been challenged under various traffic conditions with two types of applications: WAVE warning messages and On-Off applications for unicast traffic between vehicles. The unicast traffic has varied with different numbers of data flows and traffic rates.

## Simulation parameters

The performance evaluation is conducted by simulating various mobility scenarios and data traffic conditions. VANET scenarios are created of 100 nodes following the Nomadic

**Table 2 The simulated experiments parameters.**

| Parameter | Values |
| --- | --- |
| Number of nodes | 100 |
| Simulation time | 900 s |
| Simulation area | 2,500 × 500 m |
| MAC layer | IEEE 802.11p |
| Mobility model | Nomadic community |
| Group size | 10 |
| Group size deviation | 5 |
| Group distribution radius | 200 m |
| Max. reference point association time | 60 s |
| Speed limit | 50 km/h |
| Pause time | 0 s |
| Traffic applications | On/Off application, Wave application |
| WAVE application traffic interval | 0.25 s |
| WAVE application packet size | 200 bytes |
| On-Off application packet size | 128 bytes |
| On-Off application transmission rate | 2, 4, 8 Kbps |
| On-Off application number of flows | 10, 15, 20, 25, 30, 35, 40 |

mobility model described in "AODV Power-Based Light Reverse ETX (AODV-PLR)", with nodes distributed on an area size of 2,500 × 500 m, moving for a period of 900 seconds. The average group size is set to 10 nodes, with a group size variation based on the group standard deviation parameter, which is set to 5. The group distribution radius is set at 200. These settings allow group sizes to be closely around the average group size and allow the connectivity between group members to be continuous. However, the nodes themselves are mobile and move freely within this radius until the reference point changes. The association time with a reference point is set to default value of 60 seconds. Node mobility is generated using a 50 km/h speed limit, representing the regular safe road speed limit. In all the simulated scenarios, continuous movement of nodes is guaranteed by setting the pause time to 0 second. The node movement behavior is maintained during the duration of the simulated time.

These settings represent real-life VANET scenarios of moving vehicles, for example, a fleet of vehicles exploring a city, the vehicles move together from one location to another, but each vehicle explores specific areas on its own, or even a group of vehicles moving on a road and can share road information. Although it is possible to adjust the simulation settings to increase the number of nodes and simulation time, it is essential to strike a balance to maintain a manageable simulation processing time, especially with the inclusion of various evaluation dimensions. It is important to note that in order to facilitate direct and equitable comparison between the protocols, identical environmental mobility conditions are ensured with identical data traffic loads using the WAVE application as background traffic and the On-Off application for unicast traffic flows between vehicles.

The WAVE application broadcasts safety messages at specific fixed intervals. The application is set to send default safety messages of size 200 bytes every quarter of a second. On the other hand, for unicast flows, the data packets transmission rate per flow varies, including 2, 4, and 8 Kbps, with 128-byte packets. The data flows varied between 10 to 40 flows.

All obtained results represent an average of 30 runs corresponding to randomly generated topologies for each point, calculated with a 95% confidence interval with relatively small errors. Table 2 illustrates the simulation parameters used.

## EVALUATION

### Performance metrics

#### Average end-to-end delay

The packet end-to-end delay is a crucial performance metric that measures the total time a packet takes to traverse the network from source to destination. The high end-to-end delay impacts negatively on delay-sensitive applications that require real-time interaction. Optimizing delays involves reducing various aspects that affect the packet delay, including transmission, propagation, queuing, and processing delays. Therefore, an efficient routing protocol that avoids congestion is desirable to help in reducing the end-to-end delay. Keeping delays as low as possible can play a major role in supporting emerging latency-sensitive applications. The average end-to-end delay represents the average end-to-end delay for all successfully delivered packets.

#### Average throughput

Average throughput describes the end-to-end data transfer rate of data transmissions achieved over the network. It depends on various aspects, including the transmission rates, available bandwidth, interference levels, and congestion levels. Increasing throughput allows for supporting high-bandwidth applications over the network.

#### Packet loss ratio

The packet loss ratio signifies the percentage of packets undelivered to their destinations. A high packet loss degrades the application performance. The routing protocol should aim to reduce packet loss, which is affected by the operation of the routing protocol and its path selection process. One of the main reasons for packet loss related to the operation of routing protocol in wireless networks is congestion. Improving the reliability of selected routes and congestion avoidance is crucial for supporting applications over wireless networks.

#### Routing overhead

Routing overhead represents all control packets associated with the routing protocol operation that indeed consume bandwidth. Although routing is essential for building paths between communicating nodes, high and excessive control overhead reduces usable data throughput. Efficient routing protocols aim to reduce packet size and reduce the number of routing packets using various techniques to minimize the associated overhead.

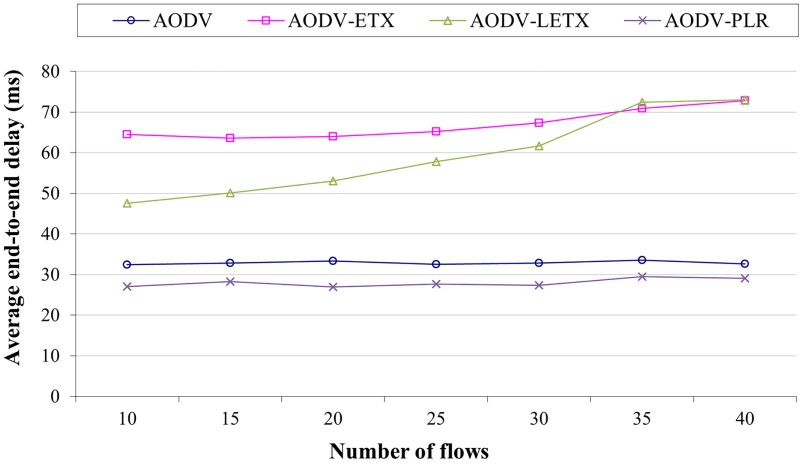

**Figure 3  Average end-to-end delay for the varied number of data flows.**

### Underutilization ratio

The underutilization ratio (UU) represents the physical transmission overhead ratio, which takes into consideration the sum of all bytes transmitted. This includes application data frames sent and the control frames associated with transmitting the data frames. A lower ratio indicates that more bandwidth is being used for application transmission. Equation (9) illustrates the calculation of this ratio where $Tx_i$ represents all transmissions in bytes, and $D\_Tx_i$ represents data transmissions in bytes.

$$UU = \left( \sum_i^n Tx_i - \sum_i^m D\_Tx_i \right) \bigg/ \sum_i^n Tx_i. \tag{9}$$

## The impact of varying numbers of flows

The impact of varying number of flows on the performance of AODV, AODV-ETX, AODV-LETX, and AODV-PLR is measured using the five performance metrics: average end-to-end delay, average throughput, packet loss, routing overhead, and underutilization ratio. The four counterparts have been evaluated under identical mobility scenarios and traffic conditions. The traffic flows have been varied to 10,15, 20, 25, 30, 35 and 40, with the traffic rate set to 2 Kbps in the presence of the WAVE application as background traffic. The speed limit is set to 50 Km/h.

Figure 3 shows the average end-to-end delay of AODV, AODV-ETX, AODV-LETX, and AODV-PLR. As the figure illustrates, almost steady behavior is demonstrated by AODV, AODV-ETX, and AODV-PLR, while the average end-to-end delay increases for AODV-LETX as the number of flows increases. The performance AODV-PLR outperforms AODV, AODV-ETX and AODV-LETX, with an average end-to-end delay that can reach around 18% lower than AODV and around 60% lower than AODV-ETX and AODV-LETX.

Figure 4 shows the average throughput for the four routing protocols. AODV-ETX has the lowest throughput across all conditions. AODV and AODV-LETX have similar

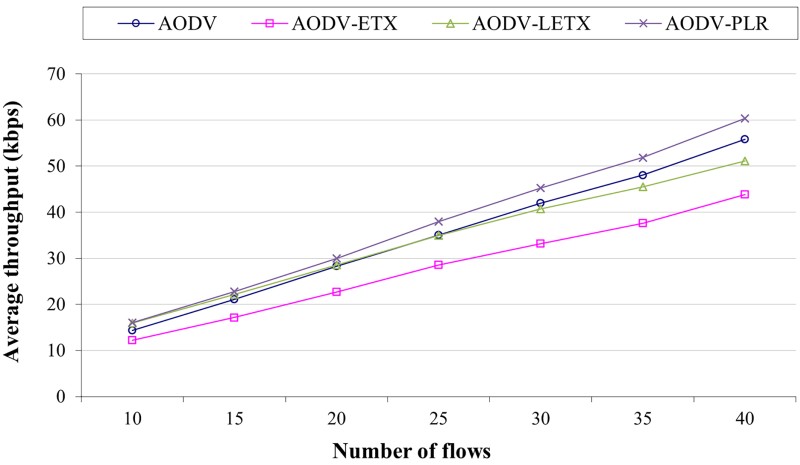

**Figure 4  Average throughput for the varied number of data flows.**

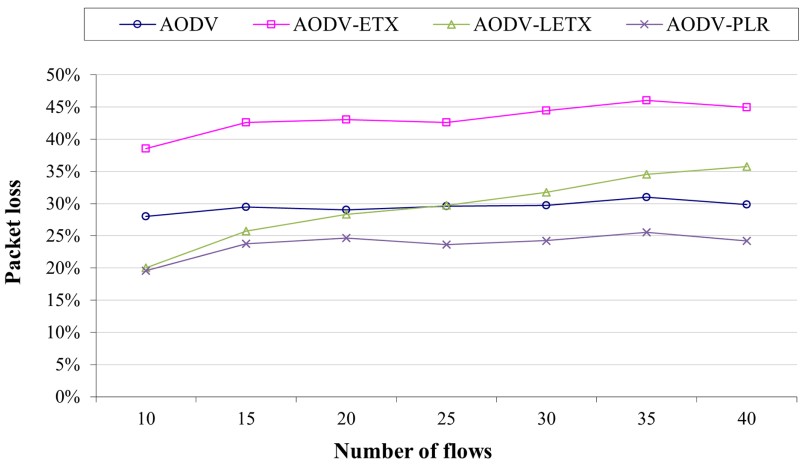

**Figure 5  Packet loss for the varied number of data flows.**

throughput in most cases, while AODV-PLR continuously outperforms the other three protocols. The performance merits of AODV-PLR highlight the advantages of incorporating the enhancements over light ETX and original ETX into the routing decisions. By considering multiple estimations, AODV-PLR uses higher-quality routes that support greater throughput. Overall, the results in Fig. 4 demonstrate that integrating additional link quality information beyond hop count can improve the performance of on-demand routing protocols like AODV. AODV-PLR outperforms AODV, AODV-ETX and AODV-LETX, with an average throughput that reaches around 8%, 18%, and 37% higher than AODV, AODV-ETX, and AODV-LETX, respectively.

Figure 5 shows AODV, AODV-ETX, AODV-LETX, and AODV-PLR packet loss. As the figure illustrates, the lowest packet loss demonstrated while testing the variation of the number of flows based on the used packet transmission rate is around 20% when 10 flows are used, demonstrated by AODV-LETX and AODV-PLR. This changes as the number of flows increases, whereas packet loss increases gradually for AODV-LETX as the number of

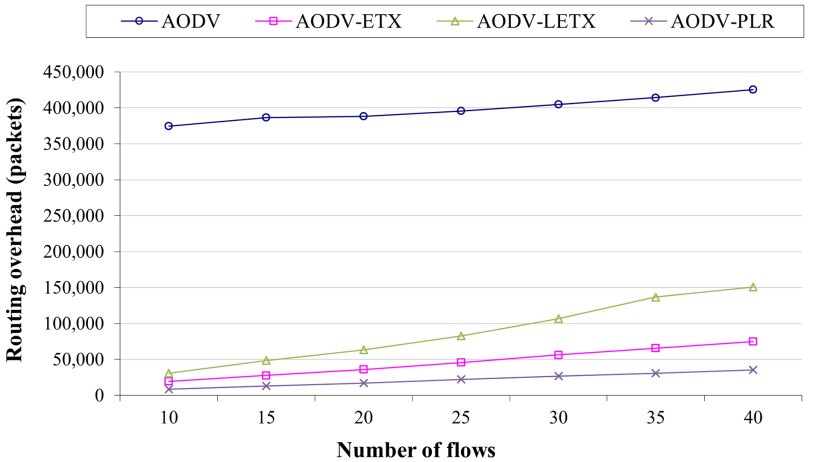

**Figure 6 Routing overhead for the varied number of data flows.**

flows increases. AODV-ETX has the highest packet loss, reaching 45% of transmitted packets. Although AODV demonstrated a stable behavior, packet loss reached 30%. In contrast, AODV-PLR demonstrated the lowest packet loss, reaching 25% of the transmitted packet, reflecting the merits of selected routes and the better route selection.

Figure 6 shows the routing overhead of the four protocols. Typically, routing overhead consumes the available bandwidth and can result in increasing congestion in the network, especially when the network is heavily loaded with data traffic. As the figure illustrates, ETX-based routing demonstrates lower overhead compared to hop count based routing represented by AODV. However, it is clear that the routing overhead increases as the number of flows increases for the four protocols due to the routing protocol effort to establish and maintain active routes between communicating nodes. The performance merits of these routing protocols can be depicted clearly as the network is populated with more data flows. AODV-PLR outperforms its counterparts with a difference that can reach 91%, 52%, and 76% lower than AODV, AODV-ETX, and AODV-LETX, respectively. This is an essential aspect of AODV-PLR as its established routes might be distributed more fairly as it embeds the RRSI influence factor in its route selection process, which adds an advantage for better network utilization.

Figure 7 depicts the underutilization ratio demonstrated by the four protocols. Lowering this ratio is a crucial aspect that represents the routing protocol's ability to utilize the network resources with reference to data transmission. As the figure illustrates, ETX-based routing demonstrates that the underutilization ratio increases with the number of flows. Although AODV has almost steady behavior, it has a high underutilization ratio of around 70%, regardless of the number of flows. However, on the other hand, ETX-based routing is affected by the increase in the number of flows, as the figure illustrates. AODV-PLR clearly outperforms its counterparts even with the increase in the number of flows, with at least 10% lower than its counterparts. This is an essential aspect of AODV-PLR as it aims in its design to reduce the routing overhead.

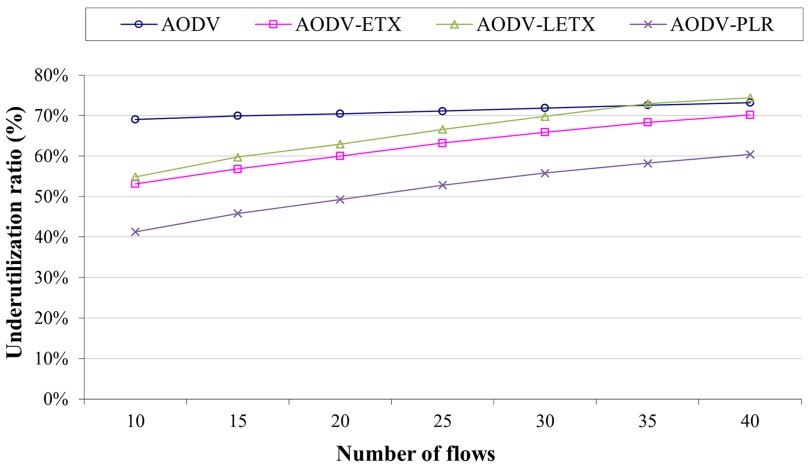

**Figure 7 Underutilization ratio for the varied number of data flows.**

## The impact of varying traffic rate

The impact of varying traffic rates on the performance of AODV, AODV-ETX, AODV-LETX, and AODV-PLR is measured using five performance metrics: average end-to-end delay, average throughput, packet loss, routing overhead, and underutilization ratio. The evaluation is conducted under identical mobility scenarios and traffic conditions. The traffic rate is varied to 2, 4, and 8 Kbps for 40 flows and the mobility speed limit of 50 Km/h in the presence of the WAVE application as background traffic.

Figure 8 shows the average end-to-end delay of the four protocols. As the figure illustrates, the end-end delay increases for AODV and AODV-LETX as the traffic rate increases and is almost steady for AODV-PLR. On the other hand, the end-to-end delay remarkably degrades for AODV-ETX with the increase in traffic rate, but this is not a reflection of the increase in traffic rate if we consider its packet loss, which reaches around 55% of the transmitted data. The performance merits of AODV-PLR are depicted clearly as the traffic increases, with an average end-to-end delay that can reach around 61%, 60%, and 61% lower than AODV, AODV-ETX and AODV-LETX, respectively.

Figure 9 shows the average throughput for the four routing protocols. As the figure illustrates, the throughput increases with traffic rate increase. AODV and AODV-PLR demonstrate close performance with a slightly higher throughput demonstrated by AODV-PLR. Regardless of the traffic rate, AODV-ETX exhibits the lowest throughput across all rates. AODV-PLR consistently outperforms ETX-based routing counterparts. This agrees with the enhancements conducted in AODV-PLR and highlights its route selection merits over light ETX and original ETX.

Figure 10 shows AODV, AODV-ETX, AODV-LETX, and AODV-PLR-ETX packet loss. As the figure illustrates, the lowest packet loss demonstrated while testing the variation of traffic rates based on the used packet transmission rate is around 25%, with the lowest traffic rate used, demonstrated by AODV-PLR. The loss ratio increases for the four protocols as the traffic rate increases, with AODV-ETX being the highest loss rate, reaching 55% of transmitted packets. The difference in packet loss between AODV and

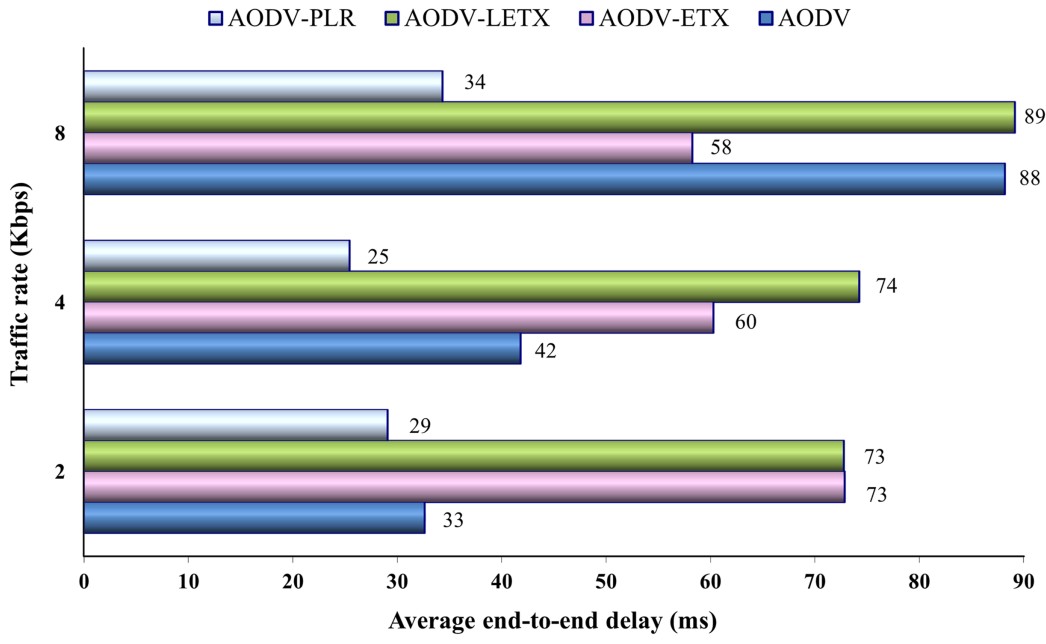

**Figure 8 Average end-to-end delay for the varied traffic rates.**

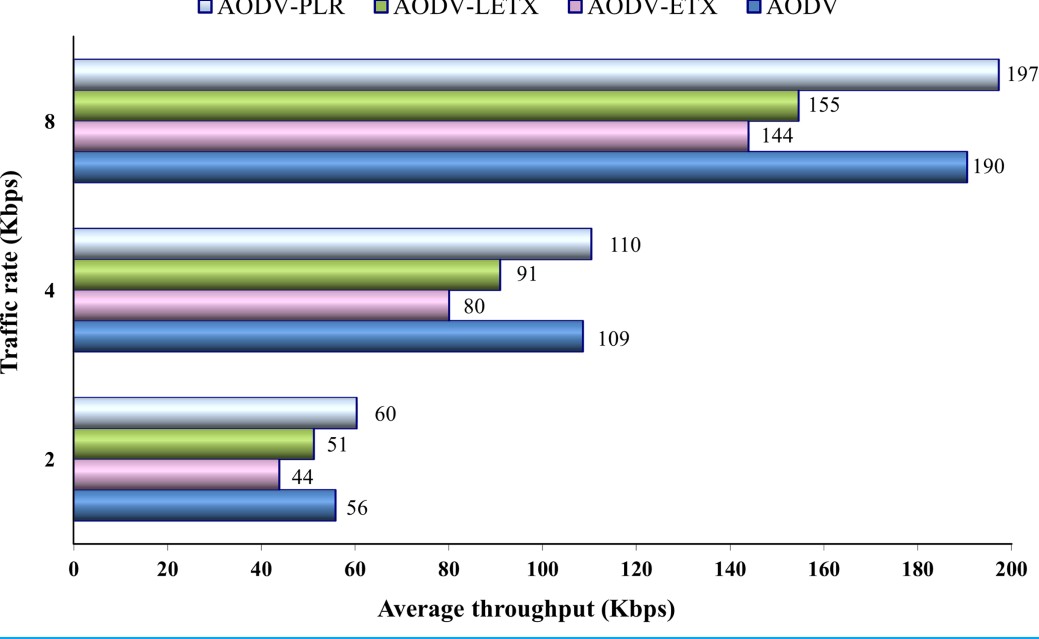

**Figure 9 Average throughput for the varied traffic rates.**

AODV-PLR decreases for data rates 4 and 8 Kbps, although AODV-PLR still shows outperformance.

Figure 11 shows the routing overhead of the four protocols. As the figure illustrates, although the traffic rate increases, the ETX-based routing demonstrates almost steady behavior in generating overhead compared to hop-count based routing represented by

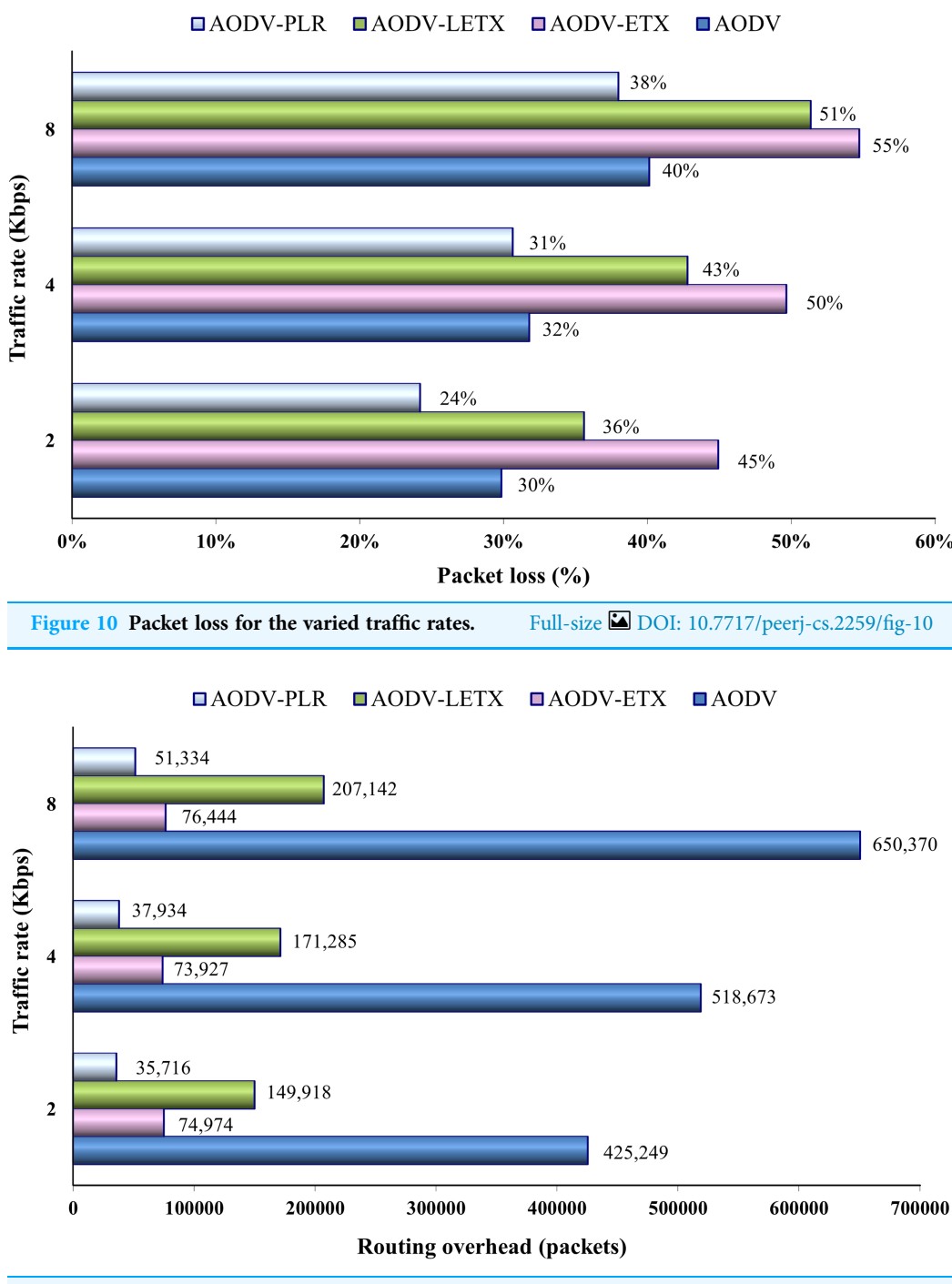

**Figure 10 Packet loss for the varied traffic rates.**

**Figure 11 Routing overhead for the varied traffic rates.**

AODV, especially in the case of AODV-ETX and AODV-PLR. The figure also depicts a sharp increase in the routing overhead generated by AODV as the traffic rate increases due to its efforts to establish and maintain active routes between communicating nodes. The performance merits of these routing protocols can be depicted clearly, with AODV-PLR outperforming its counterparts. Typically, routing overhead consumes the available

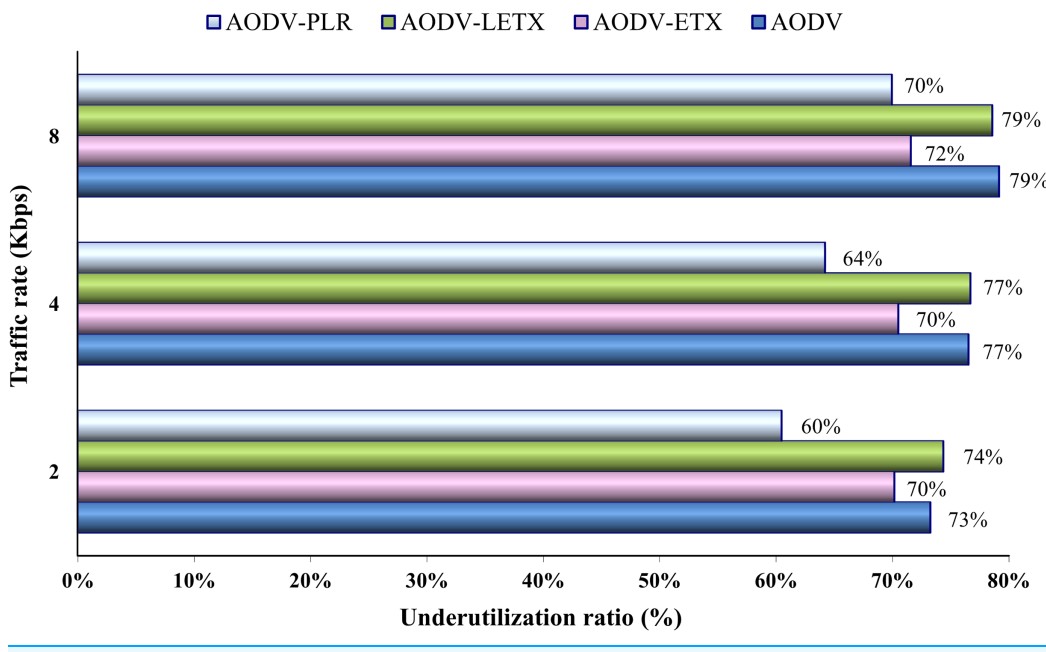

**Figure 12 Underutilization ratio for the varied traffic rates.**

bandwidth and can result in increasing congestion in the network, especially when the network is heavily loaded with data traffic.

Figure 12 depicts the underutilization ratio demonstrated by the four protocols. The impact of increasing traffic rate on this ratio is depicted clearly. As the figure illustrates, close performance is demonstrated by AODV and AODV-LETX, although their underutilization ratio increases with the traffic rate increase. Although AODV-PLR performance degrades as the traffic rate increases and the underutilization ratio increases, it still outperforms its counterparts. AODV-PLR outperforms its counterparts even at the highest traffic. However, it is important to note that the metric considers transmitted data but not necessarily delivered as an essential aspect, as it aims to measure the routing overhead relative to the amount of transmitted data.

## CONCLUSION AND FUTURE WORK

This research presented a comprehensive comparative evaluation of diverse ETX-based routing protocols for VANETs. The analysis encompassed the AODV routing protocol and three ETX-optimized variants, AODV-ETX, AODV-LETX, and AODV-PLR, under varied network traffic conditions and experienced mobility under the nomadic community mobility model. The findings show that ETX-optimized routing can provide significant performance enhancements in terms of end-to-end delay, throughput, routing overhead, packet loss and underutilization ratio. The extensive simulations demonstrated that AODV-PLR routing protocol outperforms the foundational AODV routing protocol and other enhancements, including AODV-ETX and AODV-LETX routing protocols across the performance metrics. Specifically, AODV-PLR showed lower average end-to-end delay, higher throughput, lower packet loss, lower routing overhead, and a reduced

network underutilization ratio compared to the other protocols. The improvements with AODV-PLR highlight the benefits of incorporating multiple link quality metrics into route selection strategies beyond just hop count or legacy ETX. This has been demonstrated by the performance of AODV-PLR and its ability to select higher quality routes that make more efficient use of network resources, evidenced mainly by the higher throughput, lower overhead and average end-to-end delay, and reduced packet loss. This research provides strong evidence that integrating multiple link quality estimates can enhance the performance of routing protocol in VANETs, which may serve as the base for improving QoS for applications in VANETs.

As part of future work, additional features can be added to AODV-PLR to improve its performance further to enhance its throughput and packet loss. In addition, further assessments can be incorporated to consider more extensive scales and heterogeneous networks with varied network densities. It would also be interesting to investigate the performance under various mobility models. Furthermore, it would be interesting to embed the PLR-ETX in popular proactive protocols, such as OLSR, since there is ongoing research to enhance its operation using ETX.

## ACKNOWLEDGEMENTS

This work was conducted while Dr. Al-Qassas was on sabbatical leave from Princess Sumaya University for Technology.

### Funding

The authors received no funding for this work.

### Competing Interests

The authors declare that they have no competing interests.

### Author Contributions

- Raad Al-Qassas conceived and designed the experiments, performed the experiments, analyzed the data, performed the computation work, prepared figures and/or tables, authored or reviewed drafts of the article, and approved the final draft.
- Malik Qasaimeh analyzed the data, prepared figures and/or tables, authored or reviewed drafts of the article, and approved the final draft.

### Data Availability

The ns-3 simulator is used to conduct the experiments is available in the Supplemental Files.

### Supplemental Information

Supplemental information for this article can be found online at http://dx.doi.org/10.7717/peerj-cs.2259#supplemental-information.

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
