# Peer review of "An empirical evaluation of link quality utilization in ETX routing for VANETs"

_PeerJ Computer Science, doi:10.7717/peerj-cs.2259_

## Round 0.1 · original submission · Minor Revisions

To enhance clarity and organization, it is recommended to revise the citation format to number referenced papers and clearly indicate these numbers within the text. Additionally, numbering the manuscript sections could improve its structure. The multi-metric evaluation approach is commendable, though the influence of experimental parameters, particularly node density, warrants further exploration to assess scalability and adaptability. While the manuscript demonstrates AODV-PLR's superior performance through simulations, varying parameters could strengthen the findings' real-world applicability. Notably, background clarity could be improved with subheadings, and specific explanations for abbreviations should be included in the abstract. Blurry figures in the results section need to be clarified. Furthermore, a comparative analysis with other SOTA routing protocols and the inclusion of additional mobility models would provide broader context and generalizability. Standardizing mathematical equations' formatting and increasing font size in figures are also recommended for better readability.

Reviewer 1 ·

Basic reporting

The manuscript effectively highlights the importance of routing protocols in VANETs, focusing on the expected transmission count (ETX) metric to enhance protocol reliability and efficiency. It discusses the critical role of ETX in evaluating link quality, essential for making informed routing decisions in the dynamic conditions of vehicular networks. The paper introduces and compares various ETX-based protocols, utilizing an evaluation model that examines five performance parameters: throughput, routing overhead, end-to-end delay, packet loss, and underutilization ratio. This analysis offers valuable insights for developing robust and adaptive ETX routing protocols for VANETs, catering to the needs of evolving intelligent transportation systems by understanding protocol performance across different network scenarios.

Recommendations for improvement:

1. Please revise the citation format by numbering each referenced paper and clearly indicating these numbers within the text (e.g., [1], [2]).
2. Number the sections of your manuscript for better organization, e.g., I. Introduction, II. Literature Review, etc.

Experimental design

The manuscript's multi-metric approach to evaluating the protocols is commendable as it provides a comprehensive view of performance impacts. However, I am concerned about the potential influence of experimental parameters on the results, particularly the number of nodes. Adjusting node density could significantly affect the outcomes and may provide deeper insight into the scalability and adaptability of the proposed protocols.

Validity of the findings

The manuscript convincingly shows through extensive simulations that AODV-PLR outperforms AODV-ETX, AODV-LETX, and the foundational AODV protocol across various performance metrics. However, the findings are constrained to the simulation environment with fixed parameters. Exploring variable parameters could provide a more robust validation of the protocols' effectiveness in real-world scenarios.

Cite this review as

Reviewer 2 ·

Basic reporting

This manuscript investigated the link quality utilization in the expected transmission count for VANET. The language of this paper is fine and readable. The reference is sufficient. However, the introduction of the background lacks clarity in logic. The authors could add some subheadings to make the background introduction clearer. Besides, I didn't find a specific explanation for the abbreviations when they were first mentioned in the abstract, e.g., AODV, AODV-ETX, AODV-LETX, and AODV-PLR.

Experimental design

Some figures in the experimental results are blurry, e.g., Fig.3-7.

Validity of the findings

In conclusion, this manuscript presents a comprehensive evaluation of the existing ETX routing protocol, making a certain contribution to this field at some level.

Additional comments

none

Cite this review as

Reviewer 3 ·

Basic reporting

The manuscript presents a comprehensive evaluation of ETX-based routing protocols for VANETs. However, the study would benefit from a comparative analysis with other state-of-the-art (SOTA) routing protocols to provide a broader context for the findings.

Experimental design

The experiments conducted in this research are based on simulations, utilizing only the Nomadic Community Mobility Model. The motivation behind this choice is not adequately justified. To enhance the validity and generalizability of the results, it is recommended that the authors include additional mobility models in their evaluation.

Validity of the findings

The authors are encouraged to provide a detailed analysis of the strengths and weaknesses of each protocol, culminating in a comprehensive comparison that highlights the relative advantages and disadvantages of the ETX-based routing protocols examined.

Additional comments

1. The formatting of the mathematical equations throughout the manuscript appears unconventional. It is advisable to revise the equations to adhere to standard formatting conventions.
2. The font size in Figure 2 is too small, making it difficult to read. It is recommended that the font size be increased to improve readability.

Cite this review as

---

## Round 0.2 · Minor Revisions

The reviewers appreciated your efforts in addressing the comments. Here are suggestions for further improvement: Enhance readability with subheadings in key sections, add a brief comparative discussion on mobility models, ensure figures are clear and consider more examples for validation, expand the discussion around Table 1 for clarity, maintain consistency in mathematical notation with brief explanations, and include a section on future research directions. These adjustments will enhance your manuscript's impact.

Reviewer 2 ·

Basic reporting

My previous comments have been addressed by the authors. I have no problems with this manuscript.

Experimental design

The authors have re-designed the experimental figures to make them clearer.

Validity of the findings

None

Additional comments

None

Cite this review as

Reviewer 3 ·

Basic reporting

I have reviewed the revised manuscript, and I appreciate the considerable efforts you have made to address the reviewers' comments. Here are some additional suggestions that could further improve the quality of your manuscript:
1. The use of blue coloring for section headers to improve readability is commendable. However, I recommend adding small subheadings within sections, especially in the background and experimental results sections, to further enhance the readability and logical flow of the manuscript.
2.Your response to the inclusion of the Nomadic Community Mobility Model is well-reasoned. However, integrating a brief comparative discussion about other potential mobility models within the manuscript could strengthen the justification and provide a clearer rationale for your choice.
3.The clarity of Figures 3-7 has been improved, which is appreciable. Nevertheless, ensure that all figures, including captions and legends, are easily readable at typical print and screen resolutions. You might also consider providing additional examples or case studies to further validate the robustness of your results.
4.The inclusion of Table 1 is a significant improvement. To enhance it further, consider expanding the discussion around the table in the text to emphasize the comparative analysis of each protocol's strengths and weaknesses. This additional context will help readers better understand the implications of your findings.
5.The revisions to the mathematical equations are noted. Please recheck to ensure that all symbols, notations, and formatting are consistent with standard mathematical conventions. Providing a brief explanation for each equation would be beneficial for readers who may not be as familiar with the technical aspects.
6.Your intention to consider larger scale and heterogeneous network studies in future assessments is a positive step. Detailing a short section on potential future work and the anticipated directions of your research would provide a clear vision for ongoing and future research endeavors.

Experimental design

seen the above

Validity of the findings

seen the above

Additional comments

seen the above

Cite this review as

---

## Round 0.3 · accepted · Accept

The revision looks good to me, and congratulations on your paper!